# Low-Complexity Lossless Coding of Asynchronous Event Sequences for Low-Power Chip Integration

**DOI:** 10.3390/s222410014

**Published:** 2022-12-19

**Authors:** Ionut Schiopu, Radu Ciprian Bilcu

**Affiliations:** Tampere Handset Camera Innovation Lab, Huawei Technologies Oy (Finland) Co., Ltd., 33720 Tampere, Finland

**Keywords:** low-power electronics, low-complexity codec, lossless compressio, event camera

## Abstract

The event sensor provides high temporal resolution and generates large amounts of raw event data. Efficient low-complexity coding solutions are required for integration into low-power event-processing chips with limited memory. In this paper, a novel lossless compression method is proposed for encoding the event data represented as asynchronous event sequences. The proposed method employs only low-complexity coding techniques so that it is suitable for hardware implementation into low-power event-processing chips. A first, novel, contribution consists of a low-complexity coding scheme which uses a decision tree to reduce the representation range of the residual error. The decision tree is formed by using a triplet threshold parameter which divides the input data range into several coding ranges arranged at concentric distances from an initial prediction, so that the residual error of the true value information is represented by using a reduced number of bits. Another novel contribution consists of an improved representation, which divides the input sequence into same-timestamp subsequences, wherein each subsequence collects the same timestamp events in ascending order of the largest dimension of the event spatial information. The proposed same-timestamp representation replaces the event timestamp information with the same-timestamp subsequence length and encodes it together with the event spatial and polarity information into a different bitstream. Another novel contribution is the random access to any time window by using additional header information. The experimental evaluation on a highly variable event density dataset demonstrates that the proposed low-complexity lossless coding method provides an average improvement of 5.49%, 11.45%, and 35.57% compared with the state-of-the-art performance-oriented lossless data compression codecs Bzip2, LZMA, and ZLIB, respectively. To our knowledge, the paper proposes the first low-complexity lossless compression method for encoding asynchronous event sequences that are suitable for hardware implementation into low-power chips.

## 1. Introduction

The recent research breakthroughs in the neuromorphic engineering domain have made possible the development of a new type of sensor, called the event camera, which is bioinspired by the human brain, as each pixel operates individually and mimics the behaviour of a separate nerve cell. In contrast to the conventional camera, in which all pixels are designed to capture the intensity of the incoming light at the same time, the event camera sensor reports only the changes of the incoming light intensity above a threshold, at any timestamp, and at any pixel position by triggering a sequence of asynchronous events (sometimes called spikes); otherwise it remains silent. Because each pixel detects and reports independently only the change in brightness, the event camera sensor proposes a new paradigm shift for capturing visual data.

The event camera provides a series of important technological advantages, such as a high temporal resolution as the asynchronous events can be triggered at a minimum timestamp distance of only 1μs ($10^{-6}\;$s), i.e., the event sensor can achieve a frame rate of up to 1 million (M) frames per second (fps). This is made possible thanks to the remarkable novel event camera feature of capturing all dynamic information without unnecessary static information (e.g., background), which is an extremely useful feature for capturing high-speed motion scenes for which the conventional camera usually fails to provide a good performance. Two types of sensors are currently available on the market: (i) the dynamic vision sensor (DVS) [1], which captures only the event modality; and (ii) the dynamic and active-pixel vision sensor (DAVIS) [2], which is comprised of a DVS sensor and an active pixel sensor (APS), i.e., it captures a sequence of conventional camera frames and their corresponding event data. The event camera sensors are now widely used in the computer vision domain, wherein the RGB and event-based solutions already provide an improved performance compared with state-of-the-art RGB-based solutions for applications such as deblurring [3], feature detection and tracking [4,5], optic flow estimation [6], 3D estimation [7], superresolution [8], interpolation [9], visual odometry [10], and many others. For more details regarding event-based applications in computer vision, please see the comprehensive literature review presented in [11]. To achieve high frame rates, the captured asynchronous event sequences reach high bit-rate levels when stored using the raw event representation of 8 bytes (B) per event provided by the event camera. Therefore, for better preprocessing of event data on low-power event-processing chips, novel low-complexity and efficient event coding solutions are required to be able to store without any information loss the acquired raw event data. In this paper, a novel low-complexity lossless compression method is proposed for efficient-memory representation of the asynchronous event sequences by employing a novel low-complexity coding scheme so that the proposed codec is suitable for hardware implementation into low-cost event signal processing (ESP) chips.

The event data compression domain is understudied whereas the sensor’s popularity continues to grow thanks to improved technical specifications offered by the latest class of event sensors. The problem was tackled in only a few articles that propose to either encode the raw asynchronous event sequences generated by the sensor with or without any information loss [12,13,14], or to first preprocess the event data from a sequence of synchronous event frames (EFs) that are finally encoded by employing a video coding standard [15,16]. The EF sequences are formed by using an event-accumulation process that consists of splitting the asynchronous event sequence into spatiotemporal neighbourhoods of time intervals, processing the events triggered in a single time interval, and then generating a single event for each pixel position in the EF. These performance-oriented coding solutions are too complex for hardware implementation in the ESP chip designed with limited memory, and may be integrated only in a system on a chip (SoC) wherein enough computation power and memory is available.

In our prior work [17,18], we proposed employing an event-accumulation process which first splits each asynchronous event sequence into spatiotemporal neighbourhoods by using different time-window values, and then generates the EF sequence by using a sum-accumulation process, whereby the events triggered in a time window are represented by a single event that is set as the sign of the event polarity sum and stored at the corresponding pixel position. In [17], we proposed a performance-oriented, context-based lossless image codec for encoding the sequence of event camera frames, in which the event spatial information and the event polarity are encoded separately by using the event map image (EMI) and the concatenated polarity vector (CPV). One can note that the lossless compression codec proposed in [17] is suitable for hardware implementation in SoC chips. In [18], we proposed a low-complexity lossless coding framework for encoding event camera frames by adapting the run-length encoding scheme and Elias coding [19] for EF coding. One can note that the low-complexity lossless compression codec proposed in [18] is suitable for hardware implementation in ESP chips. The goal of this work is to propose a novel low complexity-oriented lossless compression codec for encoding asynchronous event sequences, suitable for hardware implementation in ESP chips.

In summary, the novel contributions of this work are summarized as follows.

**(1)** 
A novel low-complexity lossless compression method for encoding raw event data represented as asynchronous event sequences, which is suitable for hardware implementation into ESP chips.**(2)** A novel low-complexity coding scheme for encoding residual errors by dividing the input range into several coding ranges arranged at concentric distances from an initial prediction.**(3)** 
A novel event sequence representation that removes the event timestamp information by dividing the input sequence into ordered same-timestamp event subsequences that can be encoded in separated bit streams.**(4)** 
A lossless event data codec that provides random access (RA) to any time window by using additional header information.

The remainder of this paper is organized as follows. Section 2 presents an overview of state-of-the-art methods. Section 3 describes the proposed low-complexity lossless coding framework. Section 4 presents the experimental evaluation of the proposed codecs. Section 5 draws the conclusions of this work.

## 2. State-of-the-Art Methods

To achieve an efficient representation of the large amount of event data, a first approach was proposed to losslessly (without any information loss) encode the asynchronous event representation. In [12], a lossless compression method is proposed by removing the redundancy of the spatial and temporal information by using three strategies: adaptive macrocube partitioning structure, the address-prior mode, and the time-prior mode. The method was extended in [13] by introducing an event sequence octree-based cube partition and a flexible intercube prediction method based on motion estimation and motion compensation. However, the coding performance of these methods (based on the spike coding strategy) remains limited.

In another approach, the asynchronous event representation is compressed by employing traditional lossless data compression methods. In [14], the authors present a coding performance comparison study of different traditionally based lossless data compression strategies when employed to encode raw event data. The study shows that traditional dictionary-based methods for data compression provide the best performance. The dictionary-based approach consists of searching for matches of data between the data to be compressed and a set of strings stored as a dictionary, in which the goal is to find the best match between the information maintained in the dictionary and the data to be compressed. One of the most well-known algorithms for lossless data compression is the Lempel-Ziv 77 (LZ77) algorithm [20], which was created by Lempel and Ziv in 1977. LZ77 iterates sequentially through the input string and stores any new match into a search buffer. The Zeta Library (ZLIB) [21], an LZ77 variant called deflation, proposed a strategy whereby the input data is divided into a sequence of blocks. The Lempel–Ziv–Markov chain algorithm (LZMA) [22] is an advanced dictionary-based codec developed by Igor Pavlov for lossless data compression, which was first used in the 7-Zip open source code. The Bzip2 algorithm is based on the well-known Burrows–Wheeler transform [23] for block sorting, which operates by applying a reversible transformation to a block of input data.

In a more recent approach [24], the authors propose to treat the asynchronous event sequence as a point cloud representation and to employ a lossless compression method based on a point cloud compression strategy. One can note that the coding performance of such a method depends on the performance of the geometry-based point cloud compression (G-PCC) algorithm used in the algorithm design.

Many of the upper-level applications prefer to consume the event data as an “intensity-like” image rather than asynchronous events sequence, wherein several event-accumulation processes are proposed [25,26,27,28,29,30] to form the EF sequence. Hence, in another approach, several methods are proposed to losslessly encode the generated EF sequence. The study in [14] was extended in [15] by proposing a time aggregation-based lossless video encoding method based on the strategy of accumulating events over a time interval by creating two event frames that count the number positive and negative polarity events, which are concatenated and encoded by the high-efficiency video coding (HEVC) standard [31]. Similarly, the coding performance depends on the performance of the video coding standard employed to encode the concatenated frames.

To further improve event data representation, another approach was proposed to encode the asynchronous event sequences by relaxing the lossless compression constraint problem and accepting information loss. In [32], the authors propose a macrocuboids partition of the raw event data, and they employ a novel spike coding framework, inspired by video coding, to encode spike segments. In [16], the authors propose a lossy coding method based on a quad-tree segmentation map derived from the adjacent intensity images. One can note that the information loss introduced by such methods might affect the performance of the upper-level applications.

## 3. Proposed Low-Complexity Lossless Coding Framework

Let us consider an event camera having a W×H pixel resolution. Any change of the incoming light intensity triggers an asynchronous event, $e_{i}=\left(x_{i},\;y_{i},\;p_{i},\;t_{i}\right),$ which stores (based on the sensors representation) the following information in 8 B of memory:spatial information $\left(x_{i},y_{i}\right),\forall x_{i}\in \left[1,\;H\right],y_{i}\in \left[1,\;W\right],$ i.e., the pixel positions where the event was triggered;polarity information $p_{i}\in \left\{-1,\;1\right\},$ where the symbol “−1” signals a decrease and symbol “1” signals an increase in the light intensity; andtimestamp $t_{i},$ the time when the event was triggered.
Hence, an asynchronous event sequence, denoted as $S_{T}={\left\{e_{i}\right\}}_{i=1,2,\ldots ,N_{e}},$ collects $N_{e}$ events triggered over a time period of Tμs. The goal of this paper is to encode $S_{T}$ by employing a novel, low-complexity lossless compression algorithm.

Figure 1 depicts the proposed low-complexity lossless coding framework scheme for encoding asynchronous event sequences. A novel sequence representation groups the same-timestamp events in subsequences and reorders them. Each same-timestamp subsequence is encoded in turn by the proposed method, called low-complexity lossless compression of asynchronous event sequences (LLC-ARES). LLC-ARES is built based on a novel coding scheme, called the triple threshold-based range partition (TTP).

Section 3.1 presents the proposed sequence representation. Section 3.2 presents the proposed low-complexity coding scheme. Section 3.3 presents the proposed method.

### 3.1. Proposed Sequence Representation

An input asynchronous event sequence, $S_{T},$ is arranged as a set of same-timestamp subsequences, $S_{T}={\left\{S^{k}\right\}}_{k=0,1,\ldots ,T-1},$ where each same-timestamp subsequence $S^{k}={\left\{e_{i}^{k}\right\}}_{i=1,2,\ldots ,N_{e}^{k}}={\left\{\left(x_{i}^{k},\;y_{i}^{k},\;p_{i}^{k}\right)\right\}}_{i=1,2,\ldots ,N_{e}^{k}}$ collects all $N_{e}^{k}$ events in $S_{T}$ triggered at the same timestamp $t^{k}.$ One can note that at the decoder side the timestamp information is recovered based on the subsequence length information, ${\left\{N_{e}^{k}\right\}}_{k=0,1,\ldots ,T-1},$ i.e., $t^{k}=k$ is set to all $N_{e}^{k}$ events. Each $S^{k}$ is ordered in the ascending order of the largest spatial information dimension, e.g., $y_{i}^{k}<y_{i+1}^{k}.$ However, if $y_{i}^{k}=y_{i+1}^{k},$ then $S^{k}$ is further ordered in the ascending order of the remaining dimension, i.e., $x_{i}^{k}<x_{i+1}^{k}.$

Figure 2 depicts the proposed sequence representation and highlights the difference between the sensor’s event-by-event (EE) order, depicted on the left side, and the same-timestamp (ST) order, depicted on the right side. Note that the EE order proposes to write to file, in turn, each event $e_{i}.$ Although the proposed ST order proposes to write to file the number of events of each same-timestamp subsequence, $N_{e}^{k}$ having the same-timestamp $t^{k},$ and, if $N_{e}^{k}>0,$ it is followed by the spatial and the event information of all same-timestamp events, i.e., ${\left\{x_{i}\right\}}_{i=1:N_{e}^{k}},{\left\{y_{i}\right\}}_{i=1:N_{e}^{k}},{\left\{p_{i}\right\}}_{i=1:N_{e}^{k}}.$ Section 4 shows that the state-of-the-art dictionary-based data compression methods provide an improved performance when the proposed ST order is employed to represent the input data compared with the EE order.

### 3.2. Proposed Triple Threshold-Based Range Partition (TTP)

For hardware implementation of the proposed event data codec into low-power event-processing chips, a novel low-complexity coding scheme is proposed. The binary representation range of the residual error is partitioned into smaller intervals selected by using a short-depth decision tree designed based on a triple threshold, $\Delta =(\delta_{1},\delta_{2},\delta_{3}).$ Hence, the input range is partitioned into several smaller coding ranges arranged at concentric distances from the initial prediction.

Let us consider the case of encoding x∈[1,H], i.e., a finite range, by using the prediction $\hat{x}$ by writing the binary representation of the residual error $\epsilon =x-\hat{x}$ on exactly $n_{\epsilon }$ bits. Because on the decoder side $n_{\epsilon }$ is unknown, the triple threshold Δ is used to create a decision tree having the role of partitioning the input range [1,H] into five types of coding ranges (see Figure 3a), where either the binary representation of ϵ is represented by using a different number of bits or the binary representation of *x* is written by using a different number of bits.

Let us denote $\Delta =\delta_{1}+\delta_{2}+\delta_{3},x_{1}=\hat{x}-\Delta ,$
$x_{2}=\hat{x}+\Delta ,$
$n_{\delta_{j}}=\left\lceil \log_{2}\delta_{j}\right\rceil ,\forall j=1,2,3,$
$n_{1}=\left\lceil \log_{2}x_{1}\right\rceil ,$ and $n_{2}=\left\lceil \log_{2}\left(H-x_{2}-1\right)\right\rceil .$ The 1st range, R1, is defined by using $\delta_{1}$ as $\left(\hat{x}-\delta_{1},\;\hat{x}+\delta_{1}\right)$ to represent any residual error $|\epsilon |<\delta_{1}$ on $n_{\delta_{1}}$ bits plus an additional bit for sign(ϵ). The 2nd range, R2, is defined by using $\delta_{2}$ to represent any residual error $\left|\epsilon \right|-\delta_{1}<\delta_{2}$ on $n_{\delta_{2}}$ bits plus a sign bit, i.e., $x\in (\hat{x}-\delta_{1}-\delta_{2},\;\hat{x}-\delta_{1}]$ for ϵ<0 and $x\in [\hat{x}+\delta_{1},\;\hat{x}+\delta_{1}+\delta_{2})$ for ϵ≥0. Similarly, the 3rd range, R3, is defined by using $\delta_{3}$ to represent any residual error $\left|\epsilon \right|-\delta_{1}-\delta_{2}<\delta_{3}$ on $n_{\delta_{3}}$ bits plus a sign bit. The 4th (R4) and 5th (R5) ranges are defined for |ϵ|≥Δ and used to represent x−1 on $n_{1}$ bits and H−x on $n_{2}$ bits, respectively.

Figure 3b depicts the decision tree defined by checking the following four constraints:(c1)$b_{0}$ is set by checking |ϵ|<Δ. If true then $b_{0}=0;$ otherwise, $b_{0}=1.$(c2)If $b_{0}=0,$ then $b_{1}$ is set by checking $|\epsilon |<\delta_{1}.$ If true, then $b_{1}=0$ and R1 is employed to represent ϵ on $n_{\epsilon }=n_{\delta_{1}}+1$ bits; otherwise $b_{1}=1.$(c3)If $b_{1}=1,$ then $b_{2}$ is set by checking $|\epsilon |<\delta_{1}+\delta_{2}.$ If true then $b_{2}=0$ and R2 is employed to represent ϵ on $n_{\epsilon }=n_{\delta_{2}}+1$ bits. Otherwise, $b_{1}=1$ and R3 is used to represent ϵ on $n_{\epsilon }=n_{\delta_{3}}+1$ bits.(c4)If $b_{0}=1,$ then $b_{1}$ is set by checking $x\leq x_{1}.$ If true, then $b_{1}=0$ and R4 is employed to represent x−1 on $n_{1}$ bits. Otherwise, $b_{1}=1$ and R5 is used to represent H−x on $n_{2}$ bits.
Note that the range $\left[1,\;x_{1}\right]$ contains $x_{1}$ possible values. To fully utilize the entire set of code words (i.e., including 00⋯0 having $n_{1}$ bits length), x−1 is represented on $n_{1}$ bits.

Algorithm 1 presents the pseudocode of the basic implementation of the TTP encoding algorithm. It is employed to represent a general value *x* by using the prediction $\hat{x}$, the support range [1,H], and the triple threshold parameter, Δ, as output bitstream B, which contains the decision tree bits, followed by the binary representation of the required additional information for the corresponding coding range. Algorithm 2 presents the pseudocode of the basic implementation of the corresponding TTP decoding algorithm.
**Algorithm 1:** Encode a general *x* by using TTP
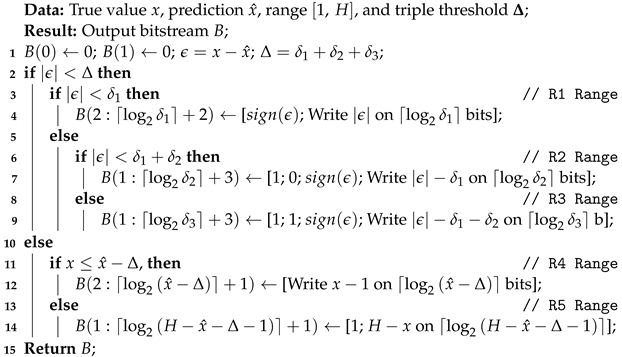


**Algorithm 2:** Decode a general *x* by using TTP

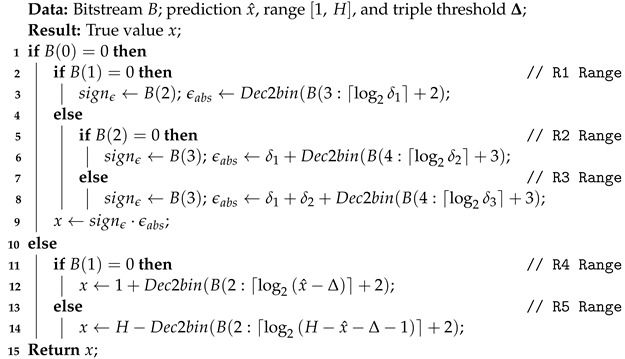



Section 3.2.1 presents the deterministic cases that may occur. Section 3.2.2 analyses the different algorithmic variations proposed to encode the data structures in the proposed event representation that have different properties.

#### 3.2.1. Deterministic Cases

In some special cases, some part of the information can be directly determined from the current coding context. For example, if $x_{1}$ or $x_{2}$ is outside the finite range (see Figure 4a), then R4 or R5 does not exist and the context tree is built without checking condition (c4), i.e., in such case one bit is saved. More exactly, steps 11–14 in Algorithms 1 and 2 are replaced with either step 12 (encode/decode using R4) or step 14 (encode/decode using R5).

Moreover, because $x_{1}$ and $x_{2}^{\prime }=H-x_{2}+1$ are not power-2 numbers, the most significant bit of $x,b_{n_{1}-1},$ is 0, thanks to the constraint $1\leq x\leq x_{1}$ and $1\leq x\leq x_{2}^{\prime },$ respectively. Figure 4b shows that if $x\in (x_{1}-2^{n_{1}-1},\;2^{n_{1}-1}]$ and $b_{n_{1}-1}$ would be set as 1, then $x>x_{1}$ and the constraint would be violated. Hence, $b_{n_{1}-1}$ is always set 0 if $x\in (x_{1}-2^{n_{1}-1},\;2^{n_{1}-1}]$, (or similarly when $x\in (x_{2}^{\prime }-2^{n_{2}^{\prime }-1},\;2^{n_{2}^{\prime }-1}]$).

#### 3.2.2. Algorithm Variations

The basic implementation of the TTP algorithm was modified for encoding different types of data. Let us denote $\epsilon_{x_{i}^{k}}=x_{i}^{k}-{\hat{x}}_{i}^{k}$ and $\epsilon_{y_{i}^{k}}=y_{i}^{k}-{\hat{y}}_{i}^{k}.$ Then the sequence ${\left\{x_{i}^{k}\right\}}_{i=1,2,\ldots ,N_{e}^{k}}$ is encoded by using version TTP${}_{x},$ where $\epsilon_{y_{i}^{k}}$ is used to detect another deterministic case: if $\epsilon_{y_{i}^{k}}=0,$ then ${\hat{x}}_{i}^{k}=x_{i-1}^{k}$ and the sign bit is saved (see Figure 2 (ST order)). The sequence ${\left\{y_{i}^{k}\right\}}_{i=2,3,\ldots ,N_{e}^{k}}$ having $\epsilon_{y_{i}^{k}}\geq 0$ (thanks to ST order) is encoded by using version TTP${}_{y},$ which is designed to encode a general value *x* found in range $[\hat{x},\;H].$
Figure 3c,d show the TTP${}_{y}$ range partitioning and decision tree, respectively.

Some data types have a very large or infinite support range. The sequence of number of events of each timestamp, ${\left\{N_{e}^{k}\right\}}_{k=0,1,\ldots ,T-1},$ is encoded by using version TTP${}_{e}.$ Note that $N_{e}^{k}\in \left[0,\;HW\right]$; however, there is a very low probability of having a large majority of pixels triggered with the same timestamp. Therefore, because $N_{e}$ is usually very small, TTP${}_{e}$ is designed to use the doublet threshold $\Delta_{e}=\left(\delta_{1},\delta_{2}\right)$, as experiments show that a triplet threshold does not improve the coding performance. Figure 3e shows the TTP${}_{e}$ range partitioning, where the values $0,1,\ldots ,\delta_{2}-2$ are encoded by R2 as the last value, $\delta_{2}-1$ (having the binary representation as $n_{\delta_{2}}$ bits of 1, i.e., $\underset{n_{\delta_{2}}}{\underset{︸}{11\ldots 1}}$), signals the use of R6 to encode |ϵ|−Δ−2 by using a simple coding technique, the Elias gamma coding (EGC) [19]. Figure 3f shows the decision tree, where $N_{e}^{k}=0$ (i.e., $S^{k}=\emptyset$) is encoded by the first bit of the decision tree.

Finally, TTP${}_{L}$ is designed to encode the length of the package bitstream $B_{\ell }$, denoted as $L_{\ell }$ (see Section 3.3.3). TTP${}_{L}$ defines seven partition intervals by using two triple thresholds: $\Delta^{S}=\left(\delta_{1}^{S},\delta_{2}^{S},\delta_{3}^{S}\right)$ is used for encoding small errors using R1S, R2S, and R3S, and $\Delta^{L}=\left(\delta_{1}^{L},\Delta_{2}^{L},\Delta_{3}^{L}\right)$ is used for encoding large errors using R1L, R2L, and R3L. Similar to TTP${}_{e},$ R6 is signalled in R3L by using the last value $\delta_{3}^{L}-1$ and $\left|\epsilon \right|-\Delta^{S}-\Delta^{L}-2$ is encoded by employing EGC [19].

### 3.3. Proposed Method

The proposed method, LLC-ARES, employs the proposed representation to generate the set of same-timestamp subsequences, ${\left\{S^{k}\right\}}_{k=0,1,\ldots ,T-1}$ (see Section 3.1). Subsequence $S^{k}$ is encoded as bitstream $B_{t^{k}}$ by using Algorithm 3, which employs the proposed coding scheme, TTP (see Section 3.2). The compressed file collects these bitstreams as $B=[B_{t^{0}}\;B_{t^{1}}\cdots B_{t^{T-1}}].$
**Algorithm 3:** Encode the subsequence of ordered events
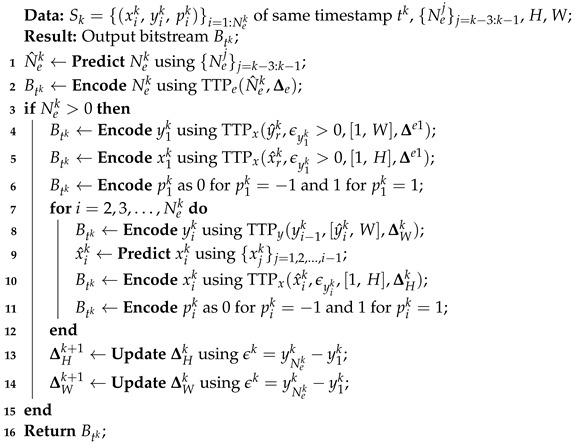


Algorithm 3 encodes the following data structures:**(i)** 
Encode $N_{e}^{k}$ by employing TTP${}_{e}$ using ${\hat{N}}_{e}^{k},$ computed by (Equation 1), and $\Delta_{e}$;**(ii)** 
Encode $e_{1}^{k}$ as follows:**(ii.1)** 
$y_{1}^{k}$ by employing TTP${}_{x}$ using ${\hat{y}}_{r}^{k}$ computed by (Equation 2), range [1,W], and $\Delta^{e1};$**(ii.2)** 
$x_{1}^{k}$ by employing TTP${}_{x}$ using ${\hat{x}}_{r}^{k}$ computed by (Equation 2), range [1,H], and $\Delta^{e1};$ and**(ii.3)** 
$p_{1}^{k}$ using binarization;**(iii)** 
The remaining events are encoded as follows:**(iii.1)** 
$y_{i}^{k}$ by employing TTP${}_{y}$ using ${\hat{y}}_{i}^{k}=y_{i-1}^{k},$ range $[{\hat{y}}_{i}^{k},\;W],$ and $\Delta_{W}^{k};$**(iii.2)** 
$x_{i}^{k}$ by employing TTP${}_{x}$ using ${\hat{x}}_{i}^{k}$ computed by (Equation 3), $\epsilon_{y_{i}^{k}},$ range [1,H], and $\Delta_{H}^{k};$ and**(iii.3)** 
$p_{i}^{k}$ using binarization.**(iv)** 
Update the triple thresholds $\Delta_{H}^{k}$ and $\Delta_{W}^{k}.$
The decoding algorithm can be simply deducted by replacing the TTP encoding algorithm in Algorithm 3 with the corresponding decoding algorithm.

Section 3.3.1 describes the prediction of each type of data used in the proposed event representation. Section 3.3.2 provides information about the setting of the triple thresholds used in the proposed method. Section 3.3.3 describes the variation of LLC-ARES algorithm to provide RA to any time window $\Delta_{RA}.$ Finally, Section 3.3.4 presents a coding example.

#### 3.3.1. Prediction

To be able to employ each one of the four algorithm variations, TTP${}_{x},$ TTP${}_{y},$ TTP${}_{e},$ and TTP${}_{L},$ four types of predictions, ${\hat{N}}_{e}^{k},\left({\hat{x}}_{r}^{k},\;{\hat{y}}_{r}^{k}\right),{\hat{x}}_{i}^{k},{\hat{L}}_{\ell },$ are computed by using the following set of equations:(1)$${\hat{N}}_{e}^{k}=\left\{\begin{array}{cc} \tau_{e} & \mathrm{if}\;k=0,\\ N_{e}^{1} & \mathrm{if}\;k=1,\\ \frac{N_{e}^{1}+N_{e}^{1}}{2} & \mathrm{if}\;k=2,\\ \frac{N_{e}^{k-3}+N_{e}^{k-2}+2N_{e}^{k-1}}{4} & \mathrm{if}\;k\geq 3. \end{array}\right.,$$
(2)$$\left({\hat{x}}_{r}^{k},\;{\hat{y}}_{r}^{k}\right)=\left\{\begin{array}{cc} \left(\frac{H}{2},\;\frac{W}{2}\right) & \mathrm{if}\;k=0,\\ \left(x_{1}^{\kappa },\;y_{1}^{\kappa }+\tau_{y}\right) & \mathrm{if}\;k>\kappa >0,N_{e}^{\kappa }>0. \end{array}\right.,$$
(3)$${\hat{x}}_{i}^{k}=\left\{\begin{array}{cc} x_{i-1}^{k} & \mathrm{if}\;i=1\;\mathrm{or}\;\epsilon_{y_{i}^{k}}=0,\\ \frac{x_{i-1}^{k}+x_{i-2}^{k}}{2} & \mathrm{if}\;i=2,\\ med({\left\{x_{i-j}\right\}}_{j=1:w_{1}}) & \mathrm{if}\;i>2\;\mathrm{and}\;|\epsilon_{y_{i}^{k}}|<\tau_{x},\\ med({\left\{x_{i-j}\right\}}_{j=1:w_{2}}) & \mathrm{if}\;i>2\;\mathrm{and}\;|\epsilon_{y_{i}^{k}}|\geq \tau_{x}. \end{array}\right.,$$
(4)$${\hat{L}}_{\ell }=\left\{\begin{array}{cc} 2^{7+\lceil log_{2}\Delta_{RA}\rceil } & \mathrm{if}\;\ell =1,\\ L_{\ell -1} & \mathrm{otherwise}. \end{array}\right..$$

In (Equation 2), the prediction for the spatial information of the first event, $e_{1}^{0},$ in the same-timestamp subsequence $S^{k},$ is set as the sensor’s centre $(\frac{H}{2},\frac{W}{2}),$ whereas the rest of the values depend on the first event $e_{1}^{\kappa }$ of the previously nonempty same-timestamp subsequence $S^{\kappa }.$ In (Equation 3), if $\epsilon_{y_{i}^{k}}$ is small, ${\hat{x}}_{i}^{k}$ is set as the median of a small prediction window of size $w_{1}$; otherwise it is of a larger prediction window of size $w_{2}.$ In our work, we set the parameters as follows: $\tau_{e}=10,$ $\tau_{x}=2^{3}+2^{4},$ $\tau_{y}=3,$ $w_{1}=5,w_{2}=15.$

#### 3.3.2. Threshold Setting

In this paper, the triple threshold parameters, $\Delta_{e},\Delta^{e1},\Delta^{S},\Delta_{H}^{k+1},\Delta_{W}^{k+1},$ and $\Delta^{L}$ are selected as power-2 numbers, and are set as follows:(5)$$\Delta_{e}=\left(2^{2},2^{2}\right),$$
(6)$$\Delta^{e1}=\left(2^{3},2^{4},2^{5}\right),$$
(7)$$\Delta_{H}^{k+1}=\left\{\begin{array}{cc} \Delta^{e1} & \mathrm{if}\;k=0\\ \left(2^{5},2^{5},2^{6}\right) & \mathrm{if}\;k>0\;\&\;\epsilon^{k}<8\\ \left(2^{4},2^{4},2^{5}\right) & \mathrm{otherwise} \end{array}\right.,$$
(8)$$\Delta_{W}^{k+1}=\left\{\begin{array}{cc} \left(2^{2},2^{3},2^{4}\right) & \mathrm{if}\;k=0\\ \left(2^{1},2^{1},2^{2}\right) & \mathrm{if}\;k>0\;\&\;\epsilon^{k}<4\\ \left(2^{1},2^{2},2^{3}\right) & \mathrm{if}\;k>0\;\&\;\epsilon^{k}<8\\ \left(2^{2},2^{2},2^{3}\right) & \mathrm{if}\;k>0\;\&\;\epsilon^{k}<16\\ \left(2^{2},2^{3},2^{4}\right) & \mathrm{otherwise} \end{array}\right.,$$
(9)$$\Delta^{S}=\left(2^{8},2^{10},2^{12}\right),$$
(10)$$\Delta^{L}=\left(2^{5+\lceil log_{2}\Delta_{RA}\rceil },2^{7+\lceil log_{2}\Delta_{RA}\rceil },2^{9+\lceil log_{2}\Delta_{RA}\rceil }\right).$$

#### 3.3.3. Random Access Functionality

LLC-ARES-RA is an LLC-ARES version which provides RA to any time window of size $\Delta_{RA}.$ Hence, $S_{T}$ is now divided into $P=\lceil \frac{T}{\Delta_{RA}}\rceil$ packages of $\Delta_{RA}$ time-length, denoted $S_{T}={\left\{S_{\ell }\right\}}_{\ell =1,2,\ldots ,P}.$ The proposed LLC-ARES is employed to encode each package $S_{\ell }$ as the bitstream set ${\left\{B_{t^{k}}\right\}}_{k=0,1,\cdots ,\Delta_{RA}-1},$ which is collected as the package *ℓ* bitstream, $B_{\ell }=\left[B_{t^{0}}\;B_{t^{1}}\;\cdots \;B_{t^{\Delta_{RA}}}\right],$ having $L_{\ell }$ bit length. The TTP${}_{L}$ version is employed to encode $L_{\ell }$ using the prediction ${\hat{L}}_{\ell },$ computed using (Equation 4), and the two triple threshold $\Delta^{S}$ and $\Delta^{L},$ and to generate the header bitstream, $B_{\ell }^{H},$ as depicted in Figure 1. Hence, the bitstreams of the set ${\left\{L_{\ell }\right\}}_{\ell =1,2,\ldots ,P}$ are collected by the header bitstream, denoted as $B^{H}=\left[B_{1}^{H}\;B_{2}^{H}\;\cdots \;B_{P}^{H}\right],$ whereas all package bitstreams are collected by the sequence bitstream, denoted as $B^{S}=\left[B_{1}\;B_{2}\;\cdots \;B_{P}\right].$ Finally, the compressed file with RA collects the $B^{H}$ and $B^{S}$ bitstreams in this order.

#### 3.3.4. A Coding Example

Figure 5 presents in detail the workflow of encoding by using the proposed LLC-ARES method an asynchronous event sequence of 2μs time-length, containing 23 triggered events. The input sequence received from the event sensor is initially represented by using the EE order. The proposed sequence representation is employed by first grouping and then rearranging the asynchronous event sequence by using the ST order. Because the input sequence contains two timestamps, the ST order consist of the same-timestamp subsequence $S^{0}$ of 10 events and the same-timestamp subsequence $S^{1}$ or 13 events. LLC-ARES encodes each data structure by using different TTP variations as described in Algorithm 3.

## 4. Experimental Evaluation

### 4.1. Experimental Setup

In our work, the experimental evaluation is carried out on large-scale outdoor stereo event camera datasets [33], called DSEC. They contain 82 asynchronous event sequences captured for network training (training data) by using the Prophesee Gen3.1 event sensor placed on top of a moving car, having a W×H=640×480 pixel resolution. All results reported in this paper use the DSEC asynchronous event sequences sorted in the ascending order of their event acquisition density. By driving at different speeds and in different outdoor scenarios, the DSEC sequences provide a highly variable density of events (see Figure 5a, in which one can see that the event density variates between 5 and 30 Mevps). Figure 6b depicts the cumulated number of events over the first 10 s of the DSEC sequences having the lowest, medium, and highest acquired event density shown in Figure 6a. To limit the runtime of state-of-the-art codecs, for each event sequence, only the first $T=10^{8}\;\mu$s (100 s) of captured event data are encoded in this work. The DSEC dataset is made publicly available online [34].

The proposed method, LLC-ARES, is implemented in the C programming language. The LLC-ARES-RA version is tested by using a time window of $\Delta_{RA}$ of $10^{2}\;\mu$s, $10^{3}\;\mu$s, and $10^{4}\;\mu$s, where for each event sequence only the first $T=10^{7}\;\mu$s of captured event data are encoded. The raw data size is computed by using the sensor specifications of 8 B per event.

The compression results are compared by using the following metrics:(c1)Compression ratio (CR), defined as the ratio between the raw data size and the compressed file size;(c2)Relative compression (RC), defined as the ratio between the compressed file size of a target codec and the compressed file size of LLC-ARES; and(c3)Bit rate (BR), defined as the ratio between the compressed file size in bits and the number of events in the asynchronous event sequence, measured in bits per event (bpev), e.g., raw data has 64 bpev.

The runtime results are compared by using the following metrics:(t1)Event density ($\rho_{E}$), defined as the ratio between the number of events in the asynchronous event sequence and the encoding/acquisition time, measured in millions of events per second (Mevps);(t2)Time ratio (TR), defined as the ratio between the data acquisition time and the codec encoding time; and(t3)Runtime, defined as the ratio between the encoding/decoding time (μs) and the number of events.

The LLC-ARES performance is compared with the following state-of-the-art traditional data compression codecs:(a)ZLIB [21] (version 1.2.3 available online [35]);(b)LZMA [22]; and(c)Bzip2 (version 1.0.5 available online [36]).
One can note that the comparison with [12] was not possible, as the codec is not publicly available and the dataset is made available only for academic research purposes.

### 4.2. Compression Results

Figure 7 shows the CR results and Figure 8 shows the BR results over DSEC [34]. One can note that, for state-of-the-art methods, the proposed ST order provides an improved performance of up to 96% compared with the sensor’s EE order. LLC-ARES (designed for low-power chip integration) provides an improved performance compared with all state-of-the-art codecs (designed for SoC integration) over the sequences having a small and medium event density, and a close performance over the sequences having a high event density as more complex coding techniques are employed by the traditional lossless data compression methods.

Table 1 shows the average CR and BR results over DSEC [34].One can note that, compared with the state-of-the-art performance-oriented lossless data compression codecs, Bzip2, LZMA, and ZLIB, the proposed LLC-ARES codec provides the following:(i)an average CR improvement of 5.49%, 11.45%, and 35.57%, respectively;(ii)an average BR improvement of 7.37%,
13.40%, and 37.12%, respectively; and(iii)an average bitsavings of 1.09 bpev, 1.99 bpev, and 5.50 bpev, respectively.

### 4.3. Runtime Results

Figure 9 shows the event density results and Figure 10 shows the TR results over DSEC. One can note that compared with runtime performance of state-of-the-art codecs, LLC-ARES provides a performance much closer to real time for all sequences, and an outstanding performance for the sequences having a high event density. More exactly, LLC-ARES provides a much faster coding speed than the state of the art for the case of high event acquisition density. Whereas the asynchronous event sequences have a very low event acquisition density, LLC-ARES provides an encoding speed as close as approximately 90% of the real-time performance (see Figure 10). Moreover, the software implementation was not optimized, as it can be further improved by a software developer expert to provide an improved runtime performance when deployed on an ESP chip.

Table 1 shows the average event density and TR results over DSEC. One can note that, compared with the state-of-the-art lossless data compression codecs, Bzip2, LZMA, and ZLIB, the proposed LLC-ARES codec provides the following:**(i)** 
an average event density improvement of 234×, 412×, and 2086×, respectively; and**(ii)** 
an average TR improvement of 216×,
401×, and 1969×, respectively.

Figure 11 and Figure 12 show the encoding and decoding runtime over DSEC, respectively. Note that LLC-ARES is a symmetric codec, wherein the encoder and decoder have similar complexity and runtime, whereas the traditional state-of-the-art lossless data compression methods are asymmetric codecs, as the encoder is much more complex than the decoder. Table 2 presents the average results over DSEC by using the EE order and the proposed ST order. Note that the LLC-ARES performance is approximately 10μs/ev for both encoding and decoding, while the traditional state-of-the-art lossless data compression methods achieve an encoding time between 135% and 515% higher than LLC-ARES and a decoding time between 92% lower and 58% higher than LLC-ARES.

The implementation of LLC-ARES was not optimized, as the implemented method must be redesigned for integration into low-power chips. These experimental results show that a proof-of-concept implementation of the algorithm on a CPU machine provides an improved performance compared with the state-of-the-art methods when tested on the same experimental setup. Please note that only LLC-ARES employs simple coding techniques so that it is suitable for hardware implementation into low-power ESP chips.

### 4.4. RA Results

Figure 13 shows the RC results over DSEC. One can note that the RC results are quite similar, as the size of the header bitstream is neglectable compared with the time-window sequence bitstream. When providing RA to the smallest tested time window of $\Delta_{RA}=100\;\mu s$, compared with LLC-ARES, the coding performance of the proposed LLC-ARES-RA method decreases with less than 0.19% when the encoded header information is stored in memory and less than 0.35% when the decoded header information is stored in memory, denoted here as memory usage (MU) results.

## 5. Conclusions

In this paper, we proposed a novel lossless compression method for encoding the event data acquired by the new event sensor and represented as an asynchronous event sequence. The proposed LLC-ARES method is built based on a novel low-complexity coding technique so that it is suitable for hardware implementation into low-power ESP chips. The proposed low-complexity coding scheme, TTP, creates short-depth decision trees to reduce either the binary representation of the residual error computed based on a simple prediction, or the binary representation of the true value. The proposed event representation employs the novel ST order, whereby same-timestamp events are first grouped into same-timestamp subsequences, and then reordered to improve the coding performance. The proposed LLC-ARES-RA method provides RA to any time window by employing a header structure to store the length of the bitstream packages.

The experimental results demonstrate that the proposed LLC-ARES codec provides an improved coding performance and a closer to real-time runtime performance compared with state-of-the-art lossless data compression codecs. More exactly, compared with Bzip2 [36], LZMA [22], and ZLIB [35], respectively, the proposed method provides:(1)an average CR improvement of 5.49%, 11.45%, and 35.57%;(2)an average BR improvement of 7.37%,
13.40%, and 37.12%;(3)an average bitsavings of 1.09 bpev, 1.99 bpev, and 5.50 bpev;(4)an average event density improvement of 234×, 412×, and 2086×; and(5)an average TR improvement of 216×,
401×, and 1969×.
To our knowledge, the paper proposes the first low-complexity lossless compression method for encoding asynchronous event sequences that is suitable for hardware implementation into low-power chips.

## Figures and Tables

**Figure 1 sensors-22-10014-f001:**
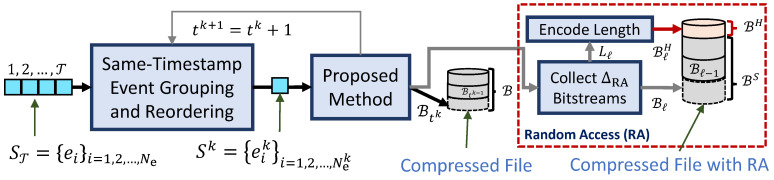
The proposed low-complexity lossless coding framework. The input asynchronous event sequence, $S_{T},$ is first represented by using the proposed event representation as a set of same-timestamp subsequences, $S^{k},$ having same-timestamp $t^{k},$ and then encoded losslessly by employing the proposed method. The output bitstream of each same-timestamp subsequence can be stored in memory as a compressed file. Moreover, it can also be collected as a package bitstream for all the timestamps found in a time period $\Delta_{RA}$ and then stored in memory together with bitstream-length information stored as a header as a compressed file with RA, so that the proposed codec can provide RA to any time window of size $\Delta_{RA}$.

**Figure 2 sensors-22-10014-f002:**
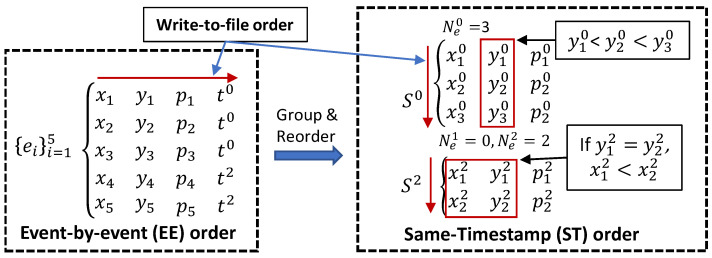
The proposed representation based the proposed same-timestamp (ST) order (on the right) in comparison with the sensor’s event-by-event (EE) order (on the left). The red arrow shows the write-to-file order used to generate the input data files feed to the traditional methods.

**Figure 3 sensors-22-10014-f003:**
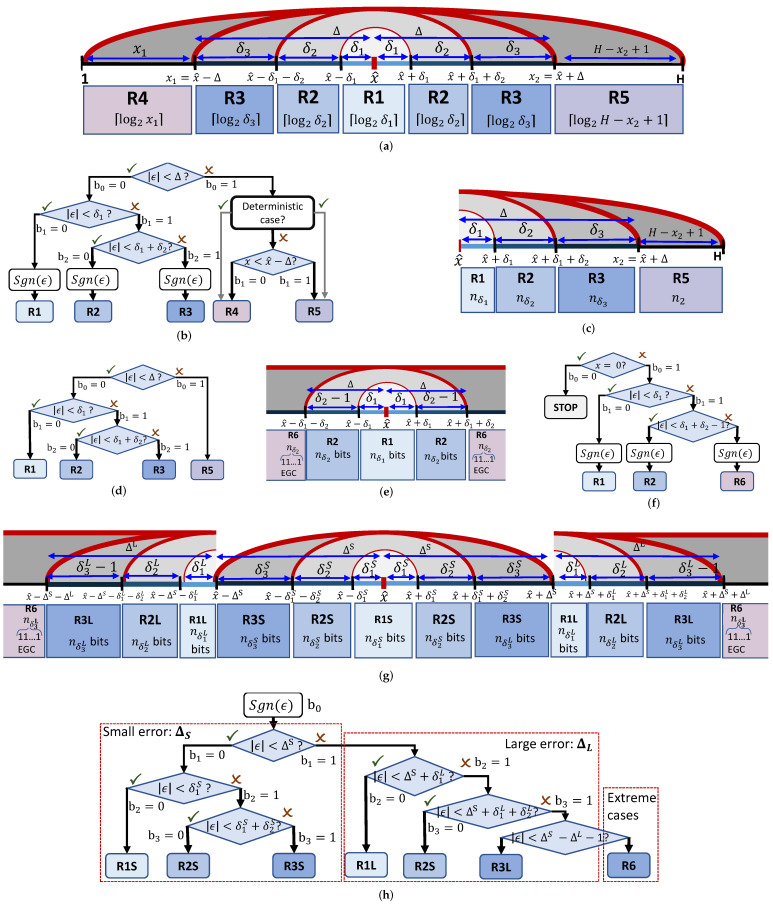
The proposed low-complexity coding scheme, triple threshold-based range partition (TTP). (**a**) TTP range partition. (**b**) TTP decision tree. (**c**) TTP${}_{y}$ range partition. (**d**) TTP${}_{y}$ decision tree. (**e**) TTP${}_{e}$ range partition. (**f**) TTP${}_{e}$ decision tree. (**g**) TTP${}_{L}$ range partition. (**h**) TTP${}_{L}$ range partition.

**Figure 4 sensors-22-10014-f004:**
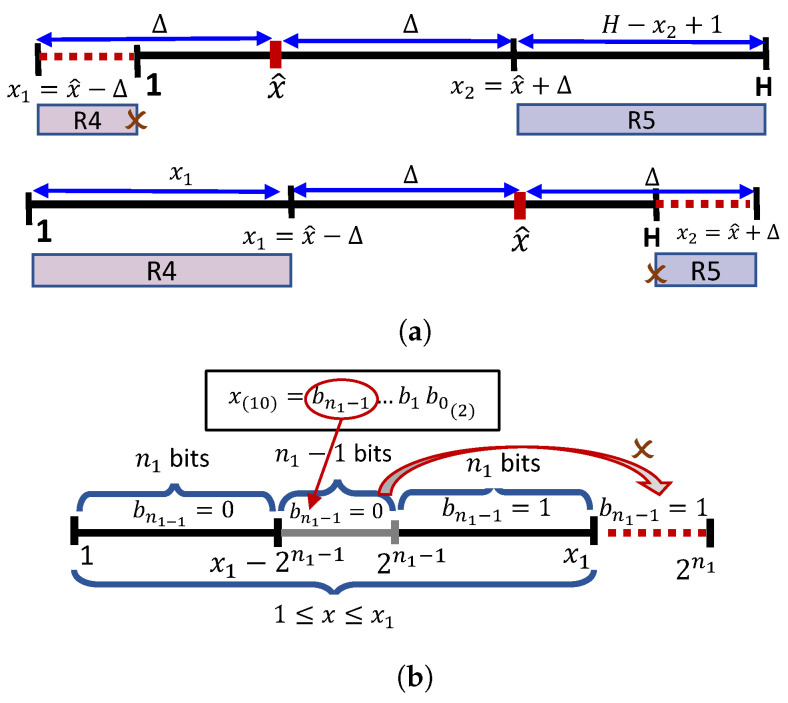
Deterministic cases: (**a**) if $x_{1}<1$ or $x_{2}>H,$ then condition (c4) is not checked when building the context tree and one bit is saved. (**b**) If $x\in (x_{1}-2^{n_{1}-1},\;2^{n_{1}-1}]$, then *x* is represented by using one bit less than in the case when $x\in [1,\;x_{1}-2^{n_{1}-1}]$ or $x\in (2^{n_{1}-1},\;x_{1}]$.

**Figure 5 sensors-22-10014-f005:**
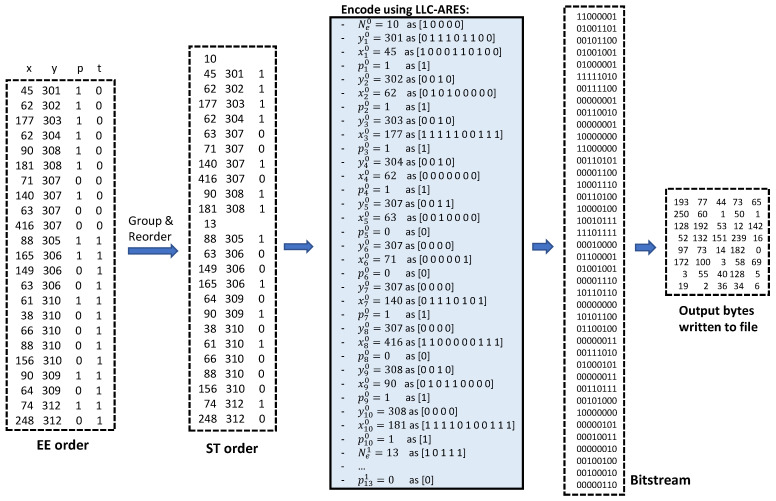
The encoding workflow using the proposed LLC-ARES method as an asynchronous event sequence of 2μs time-length, containing 23 events. The input sequence, represented by using the EE order, is first grouped and rearranged by using the proposed ST order. LLC-ARES encodes each data structure by using different TTP variations as an output bitstream of 316 bits stored by using 40 bytes, i.e., 40 numbers having an 8-bit representation.

**Figure 6 sensors-22-10014-f006:**
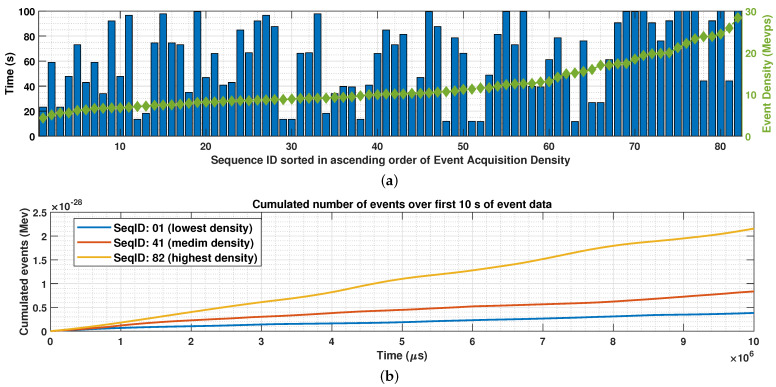
(**a**) The DSEC sequence time length (s) and event density (Mevps), where the asynchronous event sequences are sorted in ascending order of the sequence acquisition density and the sequence time length was constrained to contain only the first $T=10^{8}\;\mu$s (100 s) of the captured event data. (**b**) The cumulated number of events (Mev) over the first 10 s of the DSEC sequences having the lowest (SeqID: 01), medium (SeqID: 41), and highest (SeqID: 82) acquired event density.

**Figure 7 sensors-22-10014-f007:**
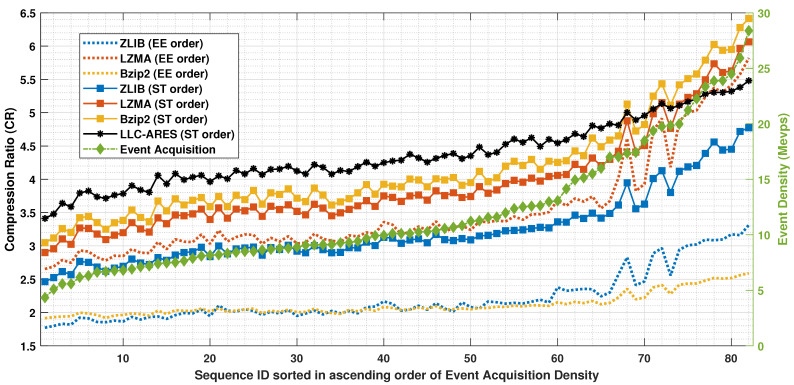
The compression ratio (CR) results over the DSEC dataset [34], where the asynchronous event sequences are sorted in ascending order of the sequence acquisition density.

**Figure 8 sensors-22-10014-f008:**
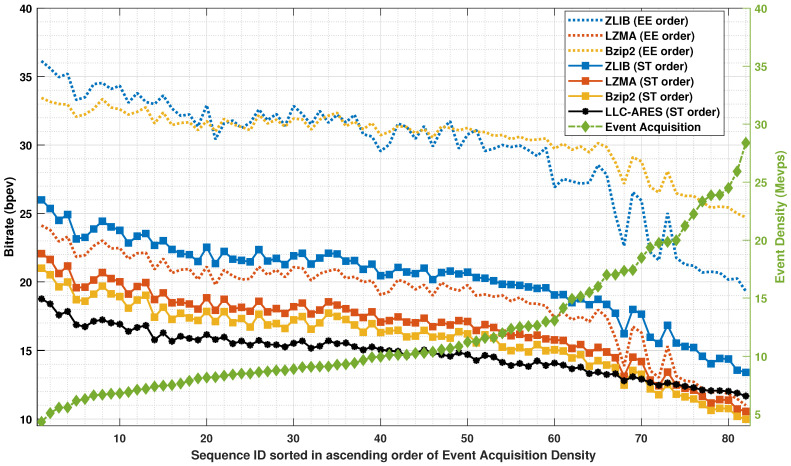
The bitrate (BR) results over DSEC [34], where the asynchronous event sequences are sorted in ascending order of the sequence acquisition density.

**Figure 9 sensors-22-10014-f009:**
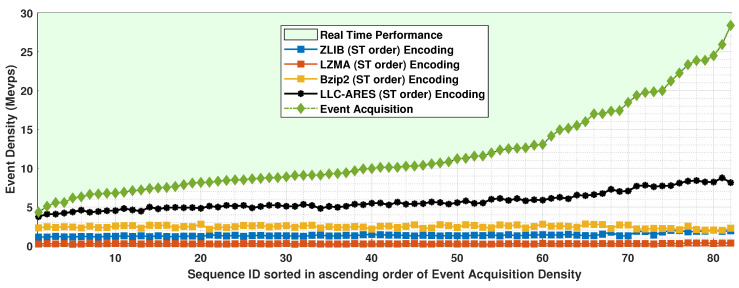
The encoded event density results over the DSEC dataset [34], where the asynchronous event sequences are sorted in ascending order of the sequence acquisition density.

**Figure 10 sensors-22-10014-f010:**
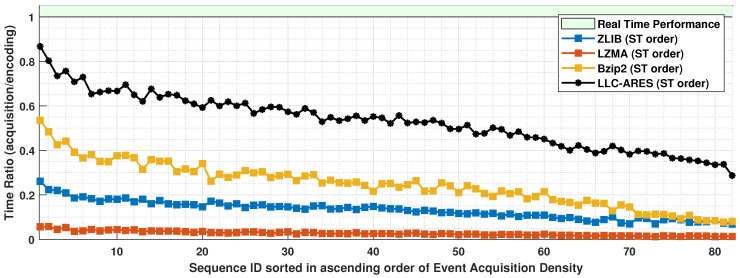
The time ratio (TR) results over the DSEC dataset [34], where the asynchronous event sequences are sorted in ascending order of the sequence acquisition density.

**Figure 11 sensors-22-10014-f011:**
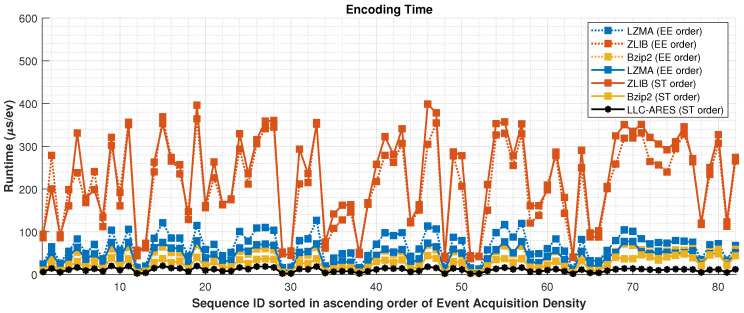
Encoding runtime results over the DSEC dataset [34], where the asynchronous event sequences are sorted in ascending order of the sequence acquisition density.

**Figure 12 sensors-22-10014-f012:**
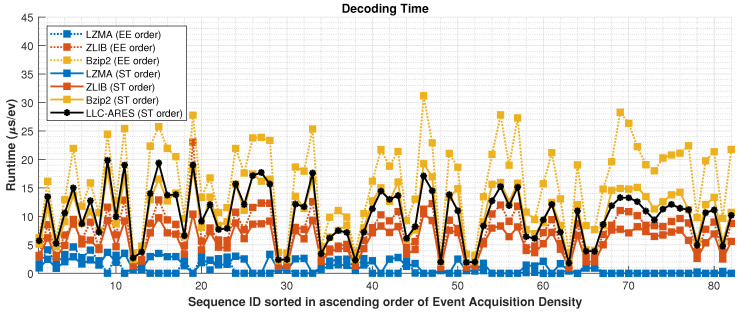
Decoding runtime results over the DSEC dataset [34], where the asynchronous event sequences are sorted in ascending order of the sequence acquisition density.

**Figure 13 sensors-22-10014-f013:**
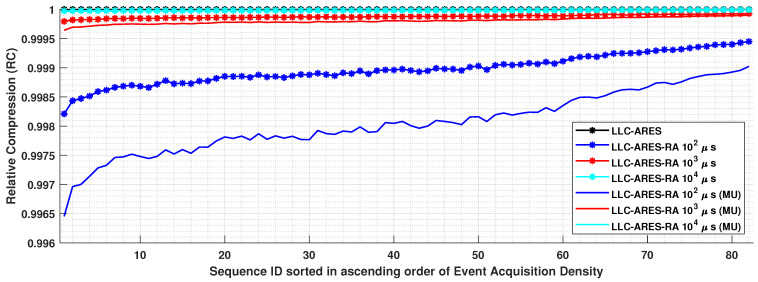
The relative compression (RC) results for RA results over the DSEC dataset [34], wherein the asynchronous event sequences are sorted in ascending order of the sequence acquisition density.

**Table 1 sensors-22-10014-t001:** Average performance over DSEC by using the EE and ST order.

Method		ZLIB [35]	LZMA [22]	bzip2 [36]	Proposed LLC-ARES
CR	EE order	2.21	3.51	2.11	–
	ST order	3.22	3.92	4.14	**4.3**
EBR (bpev)	EE order	29.65	18.91	30.50	–
	ST order	20.32	16.80	15.91	**14.8**
$\rho_{E}$ (Mevps)	ST order	1.392	0.275	2.453	**5.736**
TR	ST order	0.133	0.027	0.246	**0.531**

**Table 2 sensors-22-10014-t002:** Average runtime results over DSEC using EE and ST order.

Method		ZLIB [35]	LZMA [22]	bzip2 [36]	Proposed LLC-ARES
Encoding Runtime	EE order	67.20 μs/ev	210.39 μs/ev	40.91 μs/ev	–
	ST order	44.70 μs/ev	227.27 μs/ev	25.75 μs/ev	**10.92 μs/ev**
Decoding Runtime	EE order	**0.78 μs/ev**	7.46 μs/ev	16.09 μs/ev	–
	ST order	1.14 μs/ev	5.71 μs/ev	10.58 μs/ev	10.21 μs/ev

## Data Availability

Not applicable.

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
