# Peer review of "Low-Complexity Lossless Coding of Asynchronous Event Sequences for Low-Power Chip Integration"

_sensors, 2022, doi:10.3390/s222410014_

Round 1
Reviewer 1 Report
This paper introduces a lossless compression method for encoding the asynchronous event sequences recorded by event camera sensors. The authors take a decision tree to reduce the representation range of the residual error with less bit number. A new same-timestamp representation is also proposed.
This research topic is very important in the application areas of event camera sensors. The presentation and organization of the paper is easy to follow, and the technical details are well described. The authors provided sufficient experimental results (including comparison against other methods) to demonstrate the superiority of the proposed method. In general, I recommend to accept this paper.
Reviewer 2 Report
The authors proposed A novel lossless compression method is proposed for encoding the asynchronous event sequences acquired by event camera sensors. The proposed method employs only low-complexity coding techniques so that it is suitable for hardware implementation into low-power event-processing chips. A first contribution proposes a novel low-complexity coding scheme which uses a decision tree to reduce the representation range of the residual error. The decision tree is formed using a triplet threshold parameter which divides the input data range into several coding ranges arranged at concentric distances from an initial prediction, so that the residual error of the true value information is represented using a reduced number of bits. Another contribution proposes a novel representation by dividing the input sequence into same-timestamp subsequences, where each subsequence collects the same timestamp events in ascending order of the largest dimension of the event spatial information. The proposed same-timestamp representation replaces the event timestamp information with the
same-timestamp subsequence length and encodes it together with the event spatial and polarity information into a different bitstream. The proposed method can provide random access to any time-
window using additional header information.
1- The abstract is not prepared objectively. It should briefly highlight the paper's novelty as what is the main problem, how has it been resolved and where the novelty lies?
2- The work is interesting and I have some comments:
The paper should describes separately the problem statement, the motivation, and the real contribution.
3- In the related work section; the authors should insert the results of the previous related works and make a critical analysis by introducing the weaknesses or shortcomings of these works.
4- It would be better if the authors give an illustrative example about their proposed LLC-ARES method.
5- The authors require to show the compression and decompression time in their experimental results.
6- The author require to add a subsection to the proposed work that discuss the time and storage complexity of the proposed compression method.
7- The 'conclusions' are a key component of the paper. It should complement the 'abstract' and normally used by experts to value the paper's engineering content. In general, it should sum up the most important outcomes of the paper. It should simply provide critical facts and figures achieved in this paper for supporting the claims.
